# Monitoring changes in vitamin D levels during the COVID-19 pandemic with routinely-collected laboratory data

Lea Skapetze[1,2], Daniela Koller [1], Andreas Zwergal[3,4], Stefan Feuerriegel [5,6], Anna Rubinski [2,7] & Eva Grill [1,4,7] ✉

Vitamin D is critical for bone health and immune function, but the COVID-19 pandemic, characterized by lockdowns and reduced outdoor activity, raised concerns about potential declines in vitamin D levels due to dietary changes and limited sunlight exposure. In this study, we analyzed routinely-collected laboratory data ($N = 292,187$ patients) from a large laboratory chain in Bavaria, Germany, to assess changes in vitamin D levels before (March 2018 to February 2020) and during (March 2020 to February 2022) the pandemic. Different statistical approaches (i.e., descriptive statistics, propensity score matching, and a causal forest) were used to evaluate confounder-adjusted changes in vitamin D levels and deficiency rates. Mean vitamin D levels decreased significantly from 26.7 $\mu$g/l pre-pandemic to 26.0 $\mu$g/l during the pandemic ($p$-value < 0.001), with a corresponding increase in deficiency rates from 31.2% to 35.2% ($p$-value < 0.001). Across all statistical approaches, the decline in mean levels and the increase in deficiency rates were particularly pronounced among elderly women. These findings highlight the importance of public health strategies to monitor and improve vitamin D status, especially during periods of restricted outdoor activity.

The COVID-19 pandemic, starting in 2020, had a major global impact that extends beyond its direct effects on health. In particular, non-pharmaceutical interventions such as lockdowns restricted outdoor activities[1,2] and limited sunlight exposure, which raises concerns about potential declines in vitamin D levels. In addition, such interventions affected various aspects of well-being, including weight fluctuations due to changes in dietary behaviors[3,4]. One notable consequence has been an increase in nutrient deficiencies as a result of these dietary changes[5]. Among the nutrients relevant to maintaining overall health, vitamin D plays an important role, particularly for bone health, immune function, and inflammation[6–8]. Further, vitamin D acts as an immunomodulatory hormone, which regulates both the innate and adaptive immune systems[7,9,10]. Emerging evidence suggests that vitamin D deficiency in COVID-19 individuals was associated with prolonged mechanical ventilation and worse outcomes[11]. However, comprehensive evidence of changes in vitamin D levels during the COVID-19 pandemic is missing.

In this study, our objective was to analyze changes in vitamin D levels during the COVID-19 pandemic (see Fig. 1). For this, we used routinely-collected, real-world data (RWD) from a database[12] that was extracted from laboratory information systems. Our study comprised a large study population of $N = 292,187$ individuals from Bavaria, Germany, and spanned two distinct periods: a pre-pandemic period (March 2018–February 2020) and a pandemic

[1]Institute for Medical Information Processing, Biometry, and Epidemiology, Faculty of Medicine, LMU Munich, Munich, Germany. [2]Health Data Technologies GmbH, Neckarsulm, Germany. [3]Department of Neurology, Faculty of Medicine, LMU University Hospital, LMU Munich, Munich, Germany. [4]German Center for Vertigo and Balance Disorders (DSGZ), Faculty of Medicine, LMU University Hospital, LMU Munich, Munich, Germany. [5]LMU Munich School of Management, LMU Munich, Munich, Germany. [6]Munich Center for Machine Learning, Munich, Germany. [7]These authors contributed equally: Anna Rubinski, Eva Grill. ✉e-mail: eva.grill@med.uni-muenchen.de

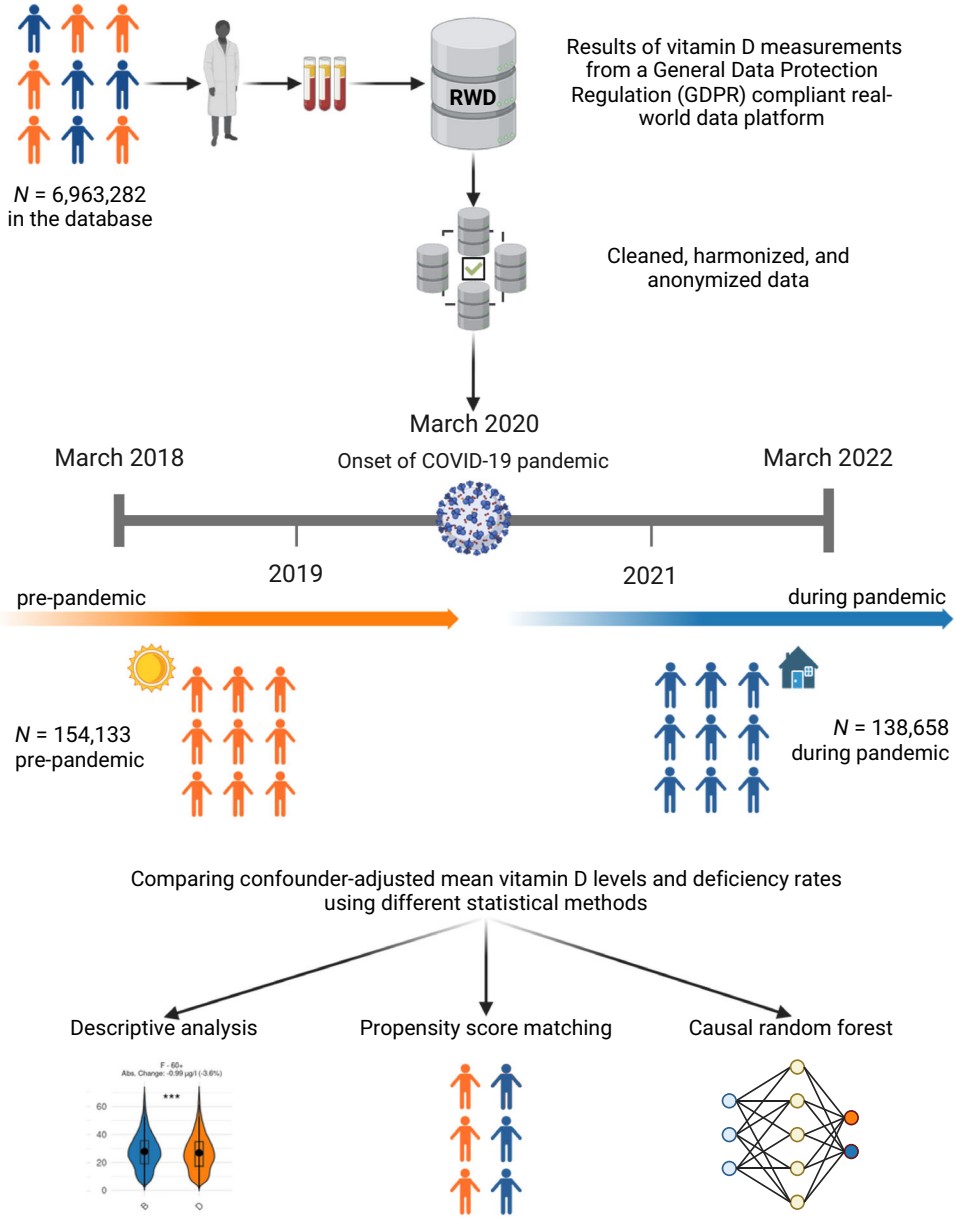

**Fig. 1 | Workflow for data collection and analysis.** This study used routinely-collected laboratory data from a large real-world data platform to compare vitamin D levels before versus during the COVID-19 pandemic. Data were extracted from laboratory information systems (LIS) to provide standardized, high-quality vitamin D measurements. We applied three different analytical approaches—namely, descriptive analysis, propensity score matching, and machine learning—to quantify confounder-adjusted changes. Results consistently indicate a significant reduction in mean vitamin D levels during the pandemic, highlighting the value of LIS-derived real-world data for public health monitoring. Created in BioRender. Skapetze, L. (2025) https://BioRender.com/ljfkj2l.

period (March 2020–February 2022). The study population reflects a comprehensive population, encompassing both inpatient and outpatient settings, individuals from all health insurance types, and a wide range of practitioner types, including general practitioners, specialists, and hospital-based physicians. The broad inclusion criteria in this study ensure that our findings are representative of real-world healthcare interactions across diverse demographic and clinical subgroups. By employing multiple methods for analysis including propensity score matching (PSM) and a causal forest model[13,14], we quantified confounder-adjusted changes in mean vitamin D levels and deficiency rates across different seasons, stratified by age and gender. Our findings highlight the value of monitoring nutrients, specifically levels of vitamin D on a large population-wide scale for public health policy.

Monitoring vitamin D levels using routinely-collected laboratory data offers several advantages. First, combining data from multiple laboratories in near real-time provides a comprehensive, population-wide assessment of vitamin D status and thus allows for granular analyses that can be stratified by sociodemographics such as age and gender. Second, by drawing from diverse population groups, RWD reduces the risk of selection bias and ensures findings have high external validity, which is critical for generalizability in clinical research[15]. Third, routinely-collected laboratory data allow for tracking vitamin D levels over time, helping to identify trends and seasonal variations. The importance and benefits of using RWD for medical research have been discussed previously[16,17], while actual studies with empirical evidence are comparatively rare. With this study, we aimed to add empirical evidence by showing the value of routinely-collected

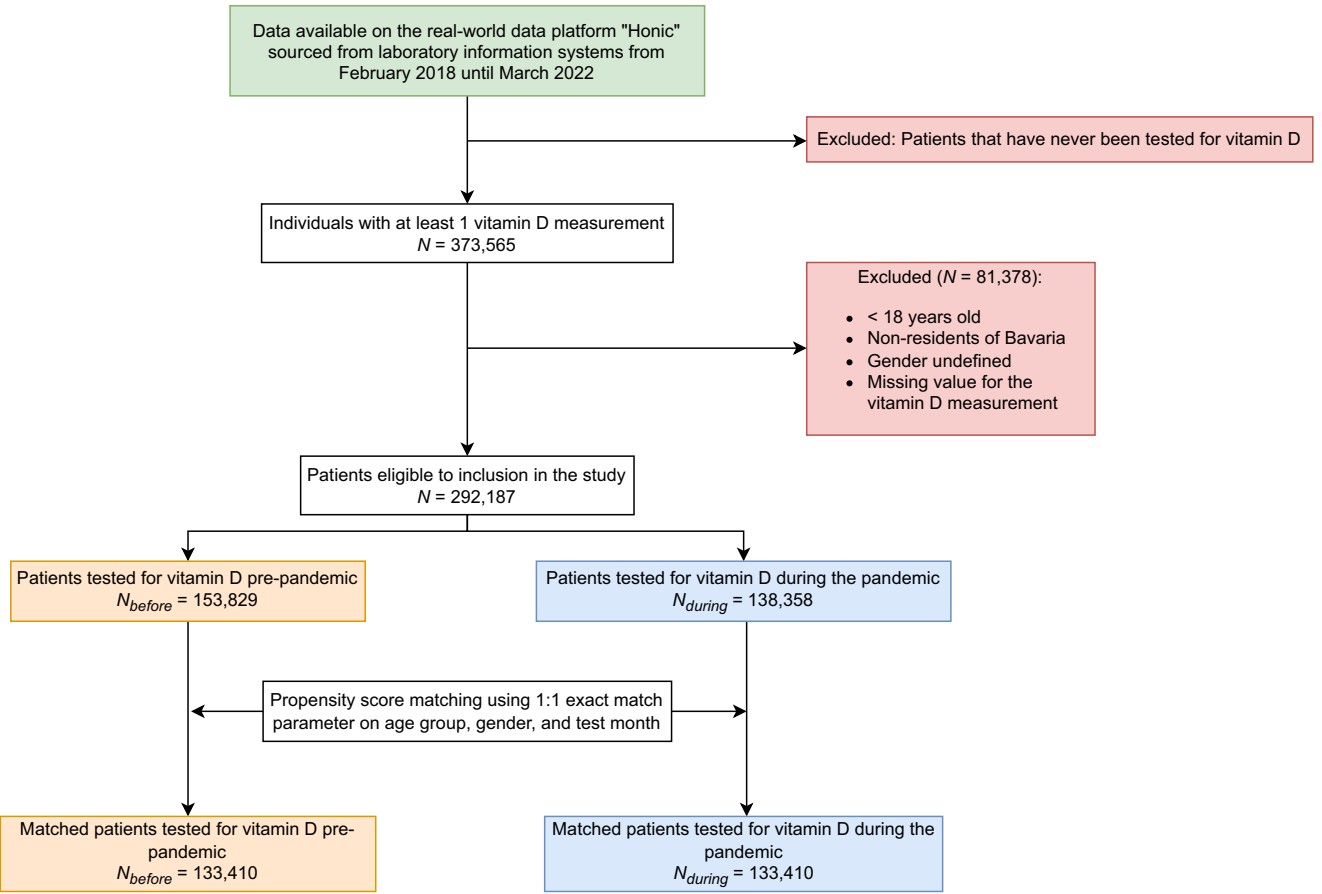

**Fig. 2 | Study population flow.** The flowchart shows the inclusion/exclusion criteria for our sample. All results are based on the sample of $N = 292,187$ individuals (the only exception is the analysis involving propensity score matching that is based on $N = 273,146$ individuals).

laboratory data for monitoring nutrient changes, in particular, vitamin D deficiencies.

## Results
### Study population
We conducted a retrospective analysis using routinely-collected laboratory data from the Honic data platform[12]. individuals were included in the current study if they underwent vitamin D testing for the first time at any point during the study period with subsequent tests excluded from the analysis (see Fig. 2). Vitamin D levels were measured as 25-Hydroxyvitamin D (25-OH-D) in serum in µg/l[18]. The final study population consisted of $N = 292,187$ individuals, which we split into the pre-pandemic period defined as March 2018 to February 2020 ($N_{pre} = 153,829$) and the pandemic period defined as March 2020 to February 2022 ($N_{during} = 138,358$) period. The definition of pre-pandemic and pandemic period follows the official declaration of the World Health Organization (WHO)[19], encompassing a total of 4 years overall with 24 months per period, providing sufficiently long observation periods of equal lengths to both account for seasonal variations and enable a robust comparison.

Individuals in the pandemic group were younger (age group 18–39 years: 27.4% vs. 22.9%) and had a slightly lower percentage of women (64.3% vs. 63.5%). Further details about Study population characteristics are presented in Table 1.

### Changes in vitamin D levels and deficiency rates during the pandemic
Differences in vitamin D health status between pre-pandemic and pandemic periods were assessed for two different outcomes, namely, changes in mean serum level and changes in deficiency rates, defined as serum levels below 20 µg/l[6]. For this, three different statistical methods were used, namely, descriptive analyses, propensity score matching, and a causal forest. Overall, the mean pre-pandemic vitamin D level was 26.7 µg/l (standard deviation [SD]: 12.8), while the mean level during the pandemic was 26.0 µg/l (SD: 13.4) (Table 2; see Supplementary Table S1 for median and interquartile range), which corresponds to a reduction of 0.7 µg/l or 2.6% ($p < 0.001$). Similarly, vitamin D deficiency rates increased significantly from 31.2% pre-pandemic to 35.2% during the pandemic ($p < 0.001$) (Table 3).

### Heterogeneity in vitamin D level changes across subpopulations
Here, we analyze heterogeneity across different age groups, genders, and all four seasons. Details are presented in Table 2 (for vitamin D levels) and Table 3 (for vitamin D deficiency rates). Differences in vitamin D levels and deficiency rates between groups were statistically tested using $t$-tests and $\chi^2$-tests, respectively.

**Age.** Vitamin D levels and deficiency rates were analyzed across three age groups, namely, 18–39, 40–59, and 60+ years. All age groups showed a consistent, statistically significant decrease in mean vitamin D levels and an increase in deficiency rates, though the magnitude of change varied by age group. During the pandemic, the age group 18–39 years had a slightly smaller decline of mean vitamin D levels (from 25.4 µg/l to 24.9 µg/l, $p < 0.001$) compared to the other age groups (40–59 years: 26.7 µg/l to 26.1 µg/l, $p < 0.001$). In contrast, the age group 18–39 years had a stronger increase in vitamin D deficiency rates (from 34.6% to 37.8%, $p < 0.001$) compared to the other age groups.

**Table 1 | Study cohort characteristics**

| | Pre-pandemic (N = 153,829) | Pandemic (N = 138,358) | p value |
|---|---|---|---|
| **Age, years (N, [%])** | | | |
| 18–39 | 35,184 [22.9] | 37,952 [27.4] | p = 2.09 × 10⁻¹⁷⁷ |
| 40–59 | 57,306 [37.3] | 50,265 [36.3] | p = 2.38 × 10⁻⁷ |
| 60+ | 61,339 [39.9] | 50,141 [36.2] | p = 1.09 × 10⁻⁹⁰ |
| **Gender (N, [%])** | | | |
| Female | 98,910 [64.3] | 87,801 [63.5] | p = 2.40 × 10⁻⁶ |
| Male | 54,919 [35.7] | 50,557 [36.5] | p = 2.40 × 10⁻⁶ |
| **Season (N, [%])** | | | |
| Winter (Dec–Feb) | 36,661 [23.8] | 36,320 [26.3] | p = 2.25 × 10⁻⁵¹ |
| Spring (Mar–May) | 44,380 [28.9] | 32,135 [23.2] | p = 3.28 × 10⁻²⁶¹ |
| Summer (Jun–Aug) | 33,474 [21.8] | 32,405 [23.4] | p = 7.78 × 10⁻²⁷ |
| Fall (Sep–Nov) | 39,314 [25.6] | 37,498 [27.1] | p = 2.70 × 10⁻²¹ |
| **Outcomes** | | | |
| Vitamin D levels, in µg/l (mean, [SD]) | 26.7 [12.8] | 26.0 [13.4] | p = 2.34 × 10⁻⁴⁷ |
| Vitamin D deficiency rate (N, [%]) | 48,056 [31.2] | 48,692 [35.2] | p = 9.84 × 10⁻¹¹⁴ |

Statistical comparisons of individual characteristics are based on two-sided Pearson $\chi^2$-tests for categorical variables and two-sided Welch's t-tests for all numerical variables. No multiple comparison adjustment.
SD standard deviation.

**Table 2 | Vitamin D levels (mean and standard deviation)**

| | Pre-pandemic | Pandemic | p-value |
|---|---|---|---|
| **Total** | 26.7 [12.8] | 26.0 [13.4] | p = 2.34 × 10⁻⁴⁷ |
| **By age, years (mean, [SD])** | | | |
| 18–39 | 25.4 [12.3] | 24.9 [12.8] | p = 7.41 × 10⁻⁹ |
| 40–59 | 26.7 [12.8] | 26.1 [13.4] | p = 2.61 × 10⁻¹⁶ |
| 60+ | 27.4 [13.1] | 26.7 [13.9] | p = 1.35 × 10⁻¹⁵ |
| **By gender (mean, [SD])** | | | |
| Females | 27.5 [13.0] | 26.7 [13.5] | p = 3.54 × 10⁻³⁷ |
| Males | 25.2 [12.4] | 24.7 [13.2] | p = 7.48 × 10⁻¹¹ |
| **By season (mean, [SD])** | | | |
| Winter (Dec – Feb) | 24.5 [13.1] | 23.8 [13.6] | p = 1.31 × 10⁻¹² |
| Spring (Mar – May) | 26.2 [13.5] | 23.9 [13.5] | p = 4.91 × 10⁻¹¹⁸ |
| Summer (Jun – Aug) | 28.6 [12.0] | 28.1 [13.1] | p = 3.76 × 10⁻⁶ |
| Fall (Sep – Nov) | 27.7 [11.8] | 28.0 [12.8] | p = 5.91 × 10⁻⁵ |

Comparison of mean vitamin D levels in serum (in µg/l) between pre-pandemic and pandemic periods across different subgroups stratified by age, gender, and season. Statistical comparisons are based on two-sided Welch's t-tests, no multiple comparison adjustment. Median and interquartile range are compared in Supplementary Table S1.
SD standard deviation.

**Gender**. Women had higher mean vitamin D levels compared to men both pre-pandemic (27.5 µg/l vs. 25.2 µg/l, p < 0.001) and during the pandemic (26.7 µg/l vs. 24.7 µg/l, p < 0.001). The decrease in mean vitamin D levels was larger for women than for men (−0.8 µg/l for women vs. −0.5 µg/l for men, p < 0.001). For men, the vitamin D deficiency rate was higher both pre-pandemic (36.1% vs. 28.5%, p < 0.001) and during the pandemic (39.8% vs. 32.6%, p < 0.001). Older

**Table 3 | Deficiency rates of vitamin D in serum**

| | Pre-pandemic | Pandemic | p value |
|---|---|---|---|
| **Total** | 48,056 [31.2] | 48,692 [35.2] | p = 9.00 × 10⁻¹¹⁴ |
| **By age, years (N, [%])** | | | |
| 18–39 | 12,179 [34.6] | 14,358 [37.8] | p = 1.57 × 10⁻¹⁹ |
| 40–59 | 17,504 [30.5] | 17,325 [34.5] | p = 7.95 × 10⁻⁴³ |
| 60+ | 18,373 [30.0] | 17,009 [33.9] | p = 1.53 × 10⁻⁴⁵ |
| **By gender (N, [%])** | | | |
| Female | 28,207 [28.5] | 28,592 [32.6] | p = 3.03 × 10⁻⁸⁰ |
| Male | 19,849 [36.1] | 20,100 [39.8] | p = 1.18 × 10⁻³³ |
| **By season (N, [%])** | | | |
| Winter (Dec – Feb) | 14,997 [40.9] | 16,066 [44.2] | p = 9.92 × 10⁻²⁰ |
| Spring (Mar – May) | 15,829 [35.7] | 14,144 [44.0] | p = 1.50 × 10⁻¹²⁰ |
| Summer (Jun – Aug) | 7352 [22.0] | 8616 [26.6] | p = 1.28 × 10⁻⁴³ |
| Fall (Sep – Nov) | 9878 [25.1] | 9866 [26.3] | p = 1.73 × 10⁻⁴ |

Deficiency rates compared before and during the pandemic across different subgroups defined by age, gender, and season. Statistical comparisons are based on two-sided Pearson's $\chi^2$-tests.
SD standard deviation.

women had the largest decrease in mean vitamin D levels and the largest increase in vitamin D deficiency rates compared to all other age/gender subgroups (see Fig. 3).

**Seasons**. To account for the natural seasonal fluctuation, the role of seasonality was analyzed additionally. A decline in mean vitamin D levels and an increase in deficiency rates were found across all seasons, yet with varying magnitudes. During the pandemic, a large decline in mean vitamin D levels was observed for spring (from 26.2 µg/l to 23.9 µg/l, p < 0.001), followed by winter (from 24.5 µg/l to 23.8 µg/l, p < 0.001). Similarly, deficiency rates were largest in spring (from 35.7 to 44.0%, p < 0.001), followed by winter (from 41.0 to 44.3%, p < 0.001). Further details are shown in Fig. 3 and Supplementary Fig. S3.

## Confounder-adjusted reductions due to the pandemic

To account for potential confounding factors such as age, gender, and seasonality, two additional methods were used (Fig. 4). First, a PSM analysis[20] was performed. By performing exact matching of individuals from the pre-pandemic and pandemic periods based on the combination of age, gender, and test month, the PSM method allows to reduce confounding such as due to selection bias and thus improve between-group comparability. The PSM-adjusted analysis showed a significant decrease in mean vitamin D levels from 26.5 µg/l (SD: 12.7) pre-pandemic to 26.0 µg/l (SD: 13.4) during the pandemic (Supplementary Table S2). This corresponds to a reduction in mean vitamin D levels of 0.5 µg/l, that is, 1.9% (p < 0.001). Vitamin D deficiency rates similarly increased from 31.6% pre-pandemic to 35.2% during the pandemic (p < 0.001) (Supplementary Table S2). The standardized mean difference (SMD) for the effect size based on the PSM analysis was −0.0388 for the confounder-adjusted reduction in mean vitamin D levels and 0.0762 for the confounder-adjusted increase in vitamin D deficiency rate (see Fig. 4).

Additionally, a causal forest model[13,14] was used as a machine learning method to estimate the confounder-adjusted impact of the pandemic on vitamin D levels. The causal forest is a state-of-the-art machine learning method for quantifying confounder-adjusted changes in an outcome variable due to intervention. In our study, we used the method to estimate the effect of the pandemic on vitamin D levels, while controlling for age, gender and season. The causal forest estimated a confounder-adjusted mean reduction in vitamin D levels of −0.678 µg/l (−2.54%) with a 99% confidence interval of −0.8019 µg/l to

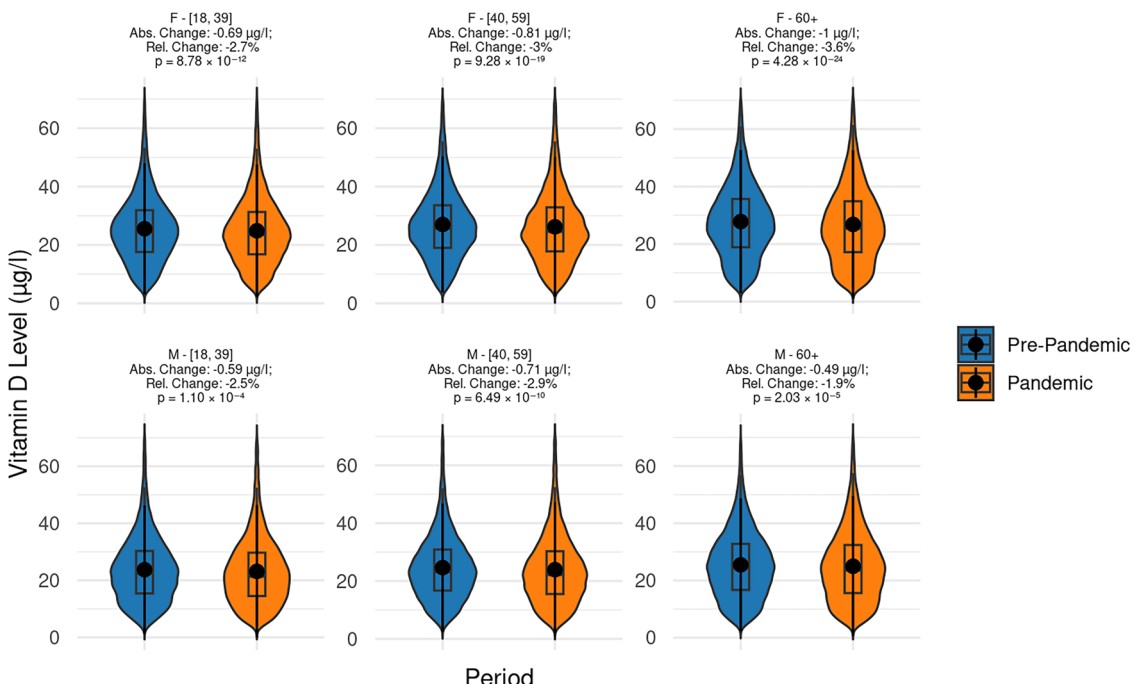

**Fig. 3 | Reduction in mean vitamin D levels during the COVID-19 pandemic across age and gender groups.** Mean vitamin D levels before (blue) and during (orange) the COVID-19 pandemic across subpopulations defined by age and gender (see captions above plot; F females, M males). Welch's unpaired two-sided *t*-tests indicate significant decreases in all groups (exact *p* values are reported), highlighting a widespread reduction in vitamin D levels during the pandemic. The absolute change represents the difference between mean vitamin D levels during the pandemic and pre-pandemic periods, while the percentage change reflects this difference as a proportion of pre-pandemic levels. $N = 292{,}187$. Center line in each box = median; box limits = 25th and 75th percentiles; whiskers = data range within $1.5 \times$ IQR; outliers plotted as individual points; violin outlines show kernel density of the distribution. Data are presented per group as distributions, not as means ± error bars. No adjustment for multiple comparisons.

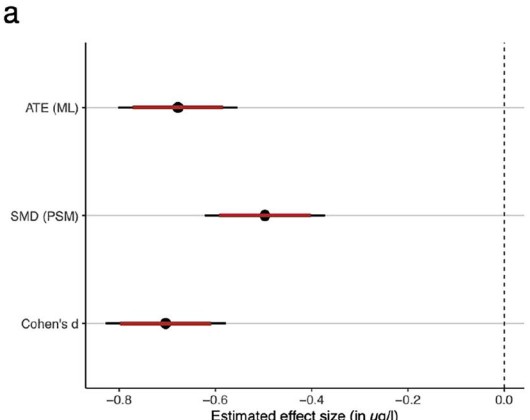
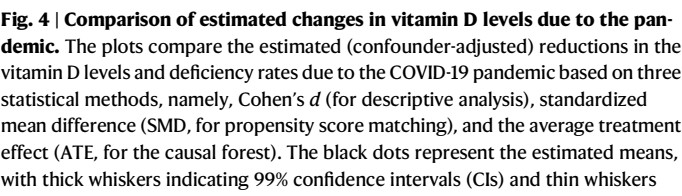
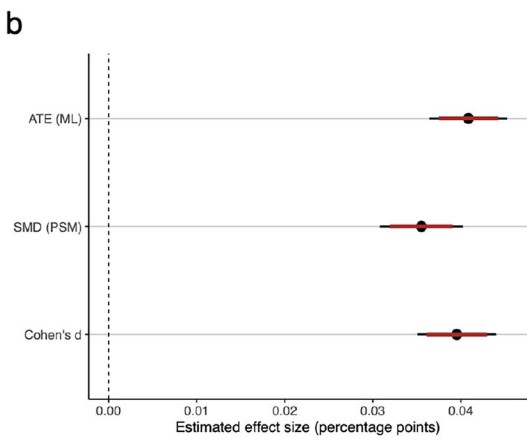

**Fig. 4 | Comparison of estimated changes in vitamin D levels due to the pandemic.** The plots compare the estimated (confounder-adjusted) reductions in the vitamin D levels and deficiency rates due to the COVID-19 pandemic based on three statistical methods, namely, Cohen's *d* (for descriptive analysis), standardized mean difference (SMD, for propensity score matching), and the average treatment effect (ATE, for the causal forest). The black dots represent the estimated means, with thick whiskers indicating 99% confidence intervals (CIs) and thin whiskers representing 95% CIs. Cohen's *d* and SMD were standardized by multiplying with the pooled standard deviation to make them comparable with the ATE from the causal forest model. Source data are provided as a Source Data file. **a** Estimated change in mean vitamin D levels. $N = 292{,}187$. Points represent mean effect estimates; error bars indicate 95% and 99% confidence intervals. **b** Estimated change in vitamin D deficiency rate. $N = 292{,}187$. Points represent mean effect estimates; error bars indicate 95% and 99% confidence intervals.

−0.554 µg/l as well as a confounder-adjusted vitamin D deficiency rate increase of 0.0408 or 4.08% with a 99% confidence interval of 0.0364 to 0.0453.

Figure 4 shows that the confidence intervals of the different methods overlap, which underscores the reliability of the confounder-adjusted estimates. Thus, all three methods applied in this study—descriptive statistics, PSM, and ML—consistently suggest a negative effect of the pandemic on vitamin D levels and a positive effect on vitamin D deficiency rates, with comparable effect sizes. We also conducted a sensitivity analysis to make sure our conclusions regarding the effect of the COVID-19 pandemic on vitamin D levels are robust to other potential sources of unobserved confounding (see

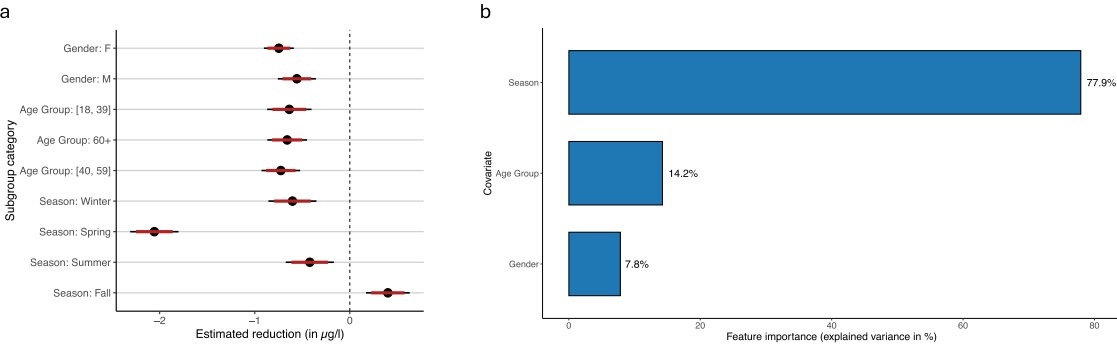

**Fig. 5 | Characterizing sources of heterogeneity.** Source data are provided as a Source Data file. **a** Subgroup analysis of estimated absolute reduction in vitamin D levels. $N = 292,187$. Points represent mean effect estimates; error bars indicate 95% and 99% confidence intervals. The plot shows the estimated absolute change with 99% (thin lines) and 95% confidence intervals (thick lines) for each subgroup category in the sensitivity analysis. For example, for the age group 40–59, a negative estimated effect of close to −1 indicates that the pandemic group had an average reduction of close to 1 µg/l in vitamin D serum levels compared to the pre-pandemic group. **b** Feature importance in predicting vitamin D levels. The plot shows the feature importance analysis reporting the explained variance due to different covariates.

Supplementary A for details), finding that it is unlikely that the reduction can be explained away by other factors.

### Predictors of heterogeneity in vitamin D reduction

To understand how the effects of the pandemic on vitamin D levels varied across different subpopulations, we analyzed the heterogeneity in the confounder-adjusted absolute reduction of vitamin D levels (Fig. 5a) by computing the confounder-adjusted reduction within different subpopulations. For example, spring exhibited the largest reduction in vitamin D levels among all seasons, while the age group 18–39 years experienced the least decline compared to older groups (see Supplementary Table S3 for further results). Figure 5b further reports a feature importance analysis from the causal forest model[21]. This analysis aimed to identify the factors that make the largest contribution to explaining variations in the reduction of vitamin D levels. As a result, feature importance analysis ranks covariates based on their contribution to the observed reduction[13]. In the causal forest model, seasonal variation emerged as the most influential factor, accounting for 77.9% of the variance in vitamin D levels, followed by the covariates representing the age groups (14.2%) and gender (7.8%).

### Robustness checks

We conducted three additional analyses as robustness checks to strengthen the public health evidence generated from monitoring population-wide vitamin D levels from routinely-collected laboratory data: (1) We performed a regression analysis to link the pandemic to changes in serum vitamin D levels and the odds of vitamin D deficiency. Here, the estimates confirm the findings from the method above (Supplementary B). (2) We retrieved the individuals' postal codes to evaluate regional variation in outcomes. Here, we specifically assessed whether outcomes vary across individuals from different degrees of urbanization (e.g., city, rural, or suburb) (Supplementary C). (3) We repeated our main analyses by replacing the seasonality dummy with a continuous time trend variable, defined as the number of months before or after the onset of the pandemic. This approach allows us to account for gradual temporal shifts rather than a discrete seasonal effect (see Supplementary D). Again, our main findings remained consistent. (4) We performed a within-subject longitudinal analysis on the subset of individuals with repeated measurements before and during the pandemic ($N = 61,393$), which revealed a statistically significant mean increase in vitamin D levels during the pandemic (Supplementary E). While our main analysis showed a population-level decrease in vitamin D levels during the pandemic, this within-subject analysis may reflect improved follow-up care or targeted interventions (e.g., Supplementaryation) in the subpopulation

of individuals with multiple measurements, and thereby provides additional nuance to the overall interpretation.

### Discussion

Understanding how the COVID-19 pandemic has affected changes in vitamin D levels is a crucial public health issue because of the long-lasting impact of vitamin D on bone health, immunity, and overall well-being[6,7]. In this study, we examined the difference in vitamin D levels across pre-pandemic (defined as March 2018 through February 2020) and pandemic (defined as March 2020 through February 2022) periods. We observed lower mean vitamin D levels and higher vitamin D deficiency rates during the pandemic as compared to pre-pandemic periods. The observed difference was robust when controlling for confounding due to age, gender, and season. Several studies have shown a decrease in vitamin D levels during the pandemic phase. However, these results pertain to specific vulnerable groups such as, for example, children[22–24], young women[25], and persons with pre-existing health problems such as chronic kidney disease[26] and previously known vitamin D deficiencies[27]. In contrast, we provide evidence from a large-scale, *population-wide* analysis that should be representative of the broader population. The observed difference was consistent across different subpopulations stratified by age and gender. Vitamin D levels fluctuated across seasons, with pronounced seasonal patterns persisting even when overall levels declined during the pandemic.

Our findings reveal surprisingly high mean serum vitamin D levels compared to previous population-based studies conducted in Germany. In our study, mean pre-pandemic vitamin D levels were 26.7 µg/l (equivalent to 66.8 nmol/l) and decreased slightly during the pandemic to 26.0 µg/l (65.0 nmol/l). These values are notably higher than those reported in the German Health Interview and Examination Survey for Adults (DEGS1)[28], which found an average level of 45.6 nmol/l between 2008 and 2011. Moreover, the DEGS1 survey reported that 61.6% of individuals had levels below the 50 nmol/l threshold, which is equivalent to the 20 µg/l threshold we used in this study. The deficiency rate reported by them is substantially higher than the 31.2% we observed pre-pandemic and the 35.2% during the pandemic. This discrepancy could be attributed to differences in sample populations and data collection methodologies. The DEGS1 survey captured a nationally representative sample, while our data originate from individuals undergoing routine laboratory testing, potentially reflecting a population more health-conscious. Additionally, increased public awareness of vitamin D supplementation and its health benefits over the past decade and, especially during the COVID-19 pandemic, may have contributed to higher levels in our study population[29].

We found that, across all age groups, the changes in mean vitamin D levels and deficiency rates during the pandemic were consistent. The mean difference was in the magnitude of 0.5 µg/l, corresponding to an absolute increase in deficiency rates of about 3–4%. Interestingly, the older age group (60+ years) exhibited higher mean vitamin D levels compared to younger adults both before and during the pandemic, a pattern contrary to expectations based on prior population-based studies[28]. This apparent paradox may be explained by differences in health behaviors and testing practices between age groups. Older adults are more likely to undergo regular health check-ups and may be more aware of vitamin D supplementation as a preventative measure for bone health and immunity. Table 1 supports this interpretation, showing a lower proportion of younger adults among those tested for vitamin D, which could reflect reduced engagement with routine healthcare services in this group (but which may be partially explained by changes in the overall health state for specific subpopulations before and after the onset of the pandemic). The COVID-19 pandemic may have further amplified these disparities, with older adults potentially prioritizing health maintenance during a time of increased health risk[30]. Additionally, the stronger increase in deficiency rates observed in the 18–39 age group (from 34.6 to 37.8%) suggests that younger adults may be more vulnerable to pandemic-related factors such as reduced outdoor activity and limited sun exposure[31]. Seasonal fluctuations in vitamin D levels, although controlled for in our analysis, may also disproportionately impact younger adults who are less likely to use supplements[32].

Our analysis revealed that both men and women experienced a decline in vitamin D levels during the pandemic. The increase in deficiency rates before and during the pandemic was similar for both genders (i.e., about 4 percent). However, men had, in general, a higher deficiency rate than women, both before and during the pandemic. This finding contrasts with the conclusions of previous reviews about vitamin D deficiency in Europe, which reported no significant sex differences in the prevalence of vitamin D deficiency across multiple studies[33]. One potential explanation for this discrepancy is behavioral differences between genders during the pandemic. Women were more likely to use vitamin D supplements than men[32], which may have contributed to their overall higher mean vitamin D levels and lower deficiency rates. Additionally, it has been demonstrated that women exhibited greater health-related worries and behavioral adjustments during the pandemic compared to men[31]. This heightened health awareness could have resulted in overall higher vitamin D levels in women compared to men. Interestingly, despite the higher serum levels and lower deficiency rates overall, the decline in levels during the pandemic was slightly larger for women than for men (−0.8 µg/l vs. −0.5 µg/l). This could be the result of the disproportionate effect of the pandemic on women's health, wealth, and social welfare, potentially exacerbating pre-existing gender inequalities and making women more vulnerable to health-related challenges during this period[34].

Seasonality explained 77.9% of the variance in COVID-19 levels (based on the feature importance analysis; Fig. 5a). During the pandemic, reduced outdoor activities, particularly during spring, likely contributed to lower vitamin D levels, as this period typically represents a time when vitamin D stores from winter are replenished through increased sun exposure[35]. Our analysis revealed that in general the most significant decline in mean vitamin D levels occurred in spring, followed by winter, as can be seen in Table 2 and in Supplementary Fig. S3. These findings are consistent with prior research documenting the seasonal variation in vitamin D levels, which tend to be lowest during late winter and early spring due to limited sunlight exposure[36].

Our data support the notion that vitamin D deficiency is more prevalent in urban populations (Supplementary C), which is consistent with earlier findings from the DEGS1 survey[28]. This pattern may be driven by environmental factors such as reduced sun exposure due to high building density, limited access to green spaces, and elevated levels of air pollution. Notably, urban air pollutants, especially tropospheric ozone, have been shown to reduce cutaneous vitamin D synthesis even in individuals with regular outdoor activity[37]. These factors may have compounded the effects of lockdowns and mobility restrictions during the COVID-19 pandemic, disproportionately affecting urban residents. These patterns suggest the need for more granular geographical analyses in future research to better understand how built environment and environmental exposures shape population wide vitamin D level.

A particular strength of our analysis is the use of RWD, which allows for large-scale, population-level insights. Data from laboratory information systems is routinely collected, providing comprehensive, near real-time data that reflects real-world health behaviors and outcomes. Our study, based on data from 292,187 individuals, captures a broad spectrum of demographic groups, enabling the exploration of how environmental and social factors-such as lockdowns and restricted outdoor activities-affected vitamin D levels during the pandemic. However, it is important to acknowledge potential discrepancies between RWD and national survey data such as those from the German National Public Health Institute (RKI)[38]. For example, RWD may be influenced by biases related to access to medical care and insurance status, while survey data are prone to response and selection biases. In the German healthcare system, vitamin D testing is often concentrated among older adults and women, who are more likely to engage in preventive health measures[39]. This may explain the overrepresentation of these groups in our dataset compared to the general population. Nevertheless, moving from impenetrable and inaccessible health data silos to health data integration is an important step for modern integrated care[40]. Our study shows the practical value of harnessing routine data from laboratory information systems. In this vein, procedural and administrative data from primary health care can increasingly be linked across services. In addition to being medically necessary, this approach also has several distinct advantages for research[41,42]. As such, RWD can be the basis of evidence for policy-makers and provide results with high external validity. A broad and longitudinal perspective enables the examination of the impact of policy changes or disruptive events on population health on a large scale.

Despite the many advantages of using routinely-collected laboratory data for monitoring population-wide vitamin D levels, our study has several limitations. First, routinely-collected data may have potential data quality concerns such as different data entry practices or inconsistencies in measurement methods. This is mitigated in our study as the data come from a leading laboratory provider with standardized measurement practices across all facilities, ensuring highly standardized and harmonized data entries. Second, the observational nature of the data limits our ability to establish direct causality between the pandemic and vitamin D deficiencies. However, we addressed this by using robust statistical methods, including confounder-adjusted estimates controlling for age, gender, and season, and by conducting a causal sensitivity analysis (see Supplementary A and Supplementary Table S4) to confirm that our findings remain consistent even in the presence of potential unobserved confounders. Third, although our data are specific to Bavaria, the demographic and healthcare characteristics of Bavaria are broadly comparable to other regions in Germany[39], suggesting that our findings may still be indicative of wider trends. Nevertheless, caution is warranted when generalizing these results to regions with different healthcare systems, climates, or population structures. Lastly, we included only male and female genders in our analysis, as entries for non-binary gender were exceedingly rare (<0.2%), which limited statistical power to make reliable inferences for this subpopulation. Also, it has to be pointed out that the gender indicated in our data is the administrative gender, which might not be necessarily identical to the biological sex of the individual.

Further, we acknowledge that as the data used in our study were derived from individuals undergoing routine laboratory testing rather than from a random sample of the general population, so there is the possibility of a shift in testing behavior over time. In particular, heightened public awareness of the possible role of vitamin D in immune function during the COVID-19 pandemic may have influenced who sought testing, potentially drawing a different subset of individuals to the laboratory than in the pre-pandemic period. This could have led to an overrepresentation of health-conscious individuals or those with particular health concerns or conditions related to vitamin D. Although we employed robust methods to adjust for observable differences in the population across time, unmeasured selection effects may nonetheless have influenced our results. We therefore explicitly acknowledge this source of bias as a potential limitation and encourage future research to further explore the impact of differential testing behavior during public health crises.

The decline in vitamin D levels during the COVID-19 pandemic has significant public health implications[43]. Targeted interventions, such as promoting supplementation and regular monitoring, are crucial for vulnerable groups, including those with limited sun exposure[27]. Our study highlights the value of using routinely-collected laboratory data for public health monitoring, enabling near real-time tracking of trends. These findings underscore the need to prioritize addressing vitamin D deficiency in public health strategies and suggest that other nutritional deficiencies may also require attention during health crises.

## Methods
### Study design
This retrospective analysis used routinely-collected laboratory data to analyze vitamin D levels in the Bavarian population before and during the COVID-19 pandemic. The primary objective was to evaluate population-wide changes in vitamin D levels by comparing the pandemic period as defined by the WHO[19] with a pre-pandemic period of equal duration. Specifically, the pre-pandemic period was defined as March 2018 to February 2020, and the pandemic period was defined as March 2020 to February 2022. Both time periods were of equal length to ensure comparability. Our analysis focused on the overall, population-wide impact of the COVID-19 pandemic on vitamin D levels, rather than examining individuals' COVID-19 infection status.

**Data source.** Data on vitamin D levels were extracted from the Honic data platform[12], which provides quality-controlled, routinely-collected healthcare data from multiple sources, including laboratory reports, outpatient visits, and prescriptions. For the current study, data primarily originated from a large laboratory chain within Bavaria comprising over 10 laboratories. Data is transmitted in real time from laboratory information systems to the Honic data platform, with approximately 12,000 to 15,000 new reports added daily. The database encompasses a highly diverse population, including individuals from both inpatient and outpatient settings, all types of health insurance, and a wide range of practitioner types, such as general practitioners, specialists, and hospital-based physicians. At the time of data extraction (03/2024), the database included over 7 million individuals, providing a comprehensive representation of real-world healthcare interactions in Bavaria.

Through the Honic platform, RWD is collected, cleaned, processed, and standardized to ensure high-quality, reliable data for analysis. The platform uses a master patient index to link patient data across sources while ensuring privacy through pseudonymization. Data access is provided within a secure analytics environment, ensuring compliance with the General Data Protection Regulation in the European Union and other data protection regulations.

**Data extraction.** All available 25-hydroxy vitamin D measurements taken during the defined time frames were extracted from the Honic data platform for this study. To maintain consistency, only the first available vitamin D measurement per individual within the study period was included, focusing on baseline levels and minimizing potential confounding due to treatment or seasonal variations from repeat measurements. Additional variables such as age, gender, postal codes, and season were also extracted.

**Inclusion criteria.** Individuals were required to have at least one vitamin D measurement within the specified time frames, as well as the following criteria: (1) *Age*: Adults 18 years and older at the time of vitamin D measurement. (2) *Residence*: Individuals residing in Bavaria, defined by postal codes beginning with 60 or ranging from 80 to 97, to ensure a regionally homogeneous sample. (3) *Administrative gender*: Male or female individuals. (4) *Vitamin D measurement*: Individuals with at least one valid value. Figure 2 shows a detailed overview of the selection criteria for the study population.

### Data processing
Prior to analysis, the extracted data was processed and anonymized to ensure data security and regulatory compliance. For this, variables that were redundant for the purpose of our analysis were removed, age was aggregated into age groups to reduce the risk of re-identification, postal codes were aggregated to degree of urbanization, and any specific identifiers were removed. As part of the latter step, postal codes (after applying the selection criteria), precise dates (retaining only month and year), and extreme outliers (that could lead to re-identification) were removed. A re-identification risk assessment was conducted on the remaining attributes, and $k$-anonymity (with $k = 5$) was applied to quasi-identifiers. Anonymization was performed using ARX version 3.9.1[44]. The final dataset was formatted as a CSV file and transferred to a secure R server for analysis.

### Measures
Vitamin D was assessed through serum levels of 25-hydroxy vitamin D (25-OH-D), the standard biomarker for evaluating vitamin D status in clinical and research settings. This biomarker is preferred because it reflects both dietary intake and endogenous production from sunlight exposure. Serum 25-OH-D concentrations were measured using immunoassays, a widely used method in routine laboratory diagnostics[18,45]. Vitamin D levels below 4 µg/L or above 300 µg/L were recorded as "<4 µg/L" or ">300 µg/L", respectively, representing the assay's detection limits. Since exact values could not be extrapolated for these cases, individuals with such recordings were excluded from the mean vitamin D levels analyses (0.4% excluded) but retained for deficiency rate calculations. According to widely accepted guidelines, a serum 25-OH-D level below 20 µg/l (50 nmol/l) was defined as vitamin D deficiency[6,46]. Outliers (values > 70 µg/l, corresponding to 0.96% of the data) were removed only for plotting to enhance visualization and prevent extreme values from distorting the scales. However, all statistical analyses, including group means and significance tests, were conducted on the full dataset to ensure the results remain representative of the population. The seasonal patterns of vitamin D levels over time are shown in Supplementary Fig. S1. A comparison between vitamin D levels during early vs. late stages of the pandemic further revealed no substantial difference (Supplementary Fig. S2).

We further use the following covariates to account for different sources of heterogeneity. (1) *Age:* The age of individuals was calculated at the measurement date and pre-grouped into three categories: 18–39 years, 40–59 years, and 60+ years. (2) *Gender:* The administrative gender was coded as male or female, individuals with unknown gender were excluded from the analysis. (3) *Season:* Seasons were categorized as follows: winter (December, January, and February), spring (March,

April, and May), summer (June, July, and August), and fall (September, October, November).

## Statistical analysis

For comparisons of continuous variables between groups (e.g., vitamin D levels before and during the pandemic), we applied two-sided *Wilcoxon rank-sum tests* (`wilcox.test` in R) when normality assumptions were not satisfied, and two-sided *t-tests* (`t.test`) otherwise. Comparisons of categorical variables (e.g., gender or age group distributions) were performed using two-sided $\chi^2$-tests (`chisq.test`). All null hypothesis tests were conducted two-sided. Effect sizes from regression models are reported as standardized or unstandardized regression coefficients ($\beta$) with corresponding confidence intervals, $p$ values, and degrees of freedom where applicable. Degrees of freedom for $t$- and $F$-statistics correspond to the residual degrees of freedom from the respective models. Confidence intervals were reported at the 99% level. A significance threshold of $\alpha = 0.01$ was applied throughout our analysis. The statistical tests allow for comparisons at the population level but do not account for potential differences in individual characteristics between the periods. Therefore, we employed two additional methods to make comparisons: (1) propensity score matching and (2) a causal forest model. These methods were specifically designed to reduce bias arising from differences in individual characteristics across the two populations and thus yield confounding-adjusted estimates.

**Propensity score matching.** PSM[20] was performed using a logistic regression model to estimate propensity scores, with the combination of age group, gender, and test month included as covariates. Nearest neighbor matching was applied, ensuring exact matches for gender, age group, and test month to create comparable pre-pandemic and pandemic groups. Test month was chosen as a covariate instead of season because it provides finer granularity in capturing potential seasonal effects and better aligns with the temporal distribution of laboratory data. After matching, the original sample of 292,187 observations was reduced to 266,820 matched observations, resulting in a loss of 25,367 records (8.7%). This matching procedure achieved strong balance across most covariates, with SMDs falling below the recommended threshold of 0.1 for the majority of variables. Two covariates exhibited slightly higher SMDs (−0.1200 for the test month of April and 0.1024 for the age group 18–39), which are marginal deviations from the threshold. These slight imbalances are unlikely to substantially affect the comparability of the matched groups[47]. By matching study populations in this manner, inferences can be drawn by comparing the intervention group (i.e., individuals tested during the pandemic) with the matched control group (i.e., individuals tested before the pandemic)[48]. See Supplementary Table S2 for more details on the population characteristics after matching. After PSM, adjusted comparisons were conducted directly within the matched sample without fitting an additional regression model. Specifically, we compared mean vitamin D levels between periods using independent sample $t$-tests and compared vitamin D deficiency rates using $\chi^2$-tests.

**Causal forest.** We further employed the causal forest model, a nonparametric state-of-the-art machine learning method[13,14,49]. The causal forest aims at estimating heterogeneous treatment effects and is thus well-suited for our task. First, causal forests share many favorable properties with random forests in that they are expressive, require little tuning, and have a low risk of overfitting, which often leads to a robust performance in practice. Second, under common mathematical assumptions, causal forests have been shown to be pointwise consistent for the true treatment effect and therefore lead to provably valid inferences[14]. Third, the causal forest makes few parametric assumptions of the form of the heterogeneous treatment effect and can thus model non-linear relationships, even in the presence of high-dimensional covariates, and typically offers better statistical power than other estimators.

The causal forest extends the traditional random forest[50] by estimating changes in an outcome variable due to an intervention, while controlling for other observed confounders (here: age, gender, and season). The causal forest proceeds by first building an ensemble of decision trees to model the relationship between intervention, outcome, and covariates. For each tree, it splits the data recursively into subgroups based on covariates, so that heterogeneity in the change of the outcome variable due to the intervention is discovered. To avoid overfitting and ensure unbiased estimates, causal forests use a technique called "honesty"[14], where one part of the data is used to determine the splits (i.e., how to group similar individuals) and where another part of the data is used for estimating the changes in the outcome variable.

The causal forest model was estimated for both outcomes of interest, namely, changes in vitamin D levels and vitamin D deficiency rates. The following covariates were used: age, gender, and season. Here, season was chosen over test month in the machine learning analysis as it captures broader environmental and behavioral patterns influencing vitamin D levels, while reducing potential noise from month-to-month variations. The causal forest model was run with the default parameter of 2000 trees[51], ensuring sufficient depth and robustness in estimating heterogeneous treatment effects. All other hyperparameters were set to their default values as in ref. 51.

To validate the ignorability assumption, we applied balance diagnostics by checking the weighted absolute standardized mean difference (ASMD) for the three key covariates (age group, gender, and test month) between the pre-pandemic and pandemic study populations. A value close to 0 indicates that the propensity scores are well-calibrated. In our analysis, all ASMD values were well below the recommended 0.1 threshold, confirming a strong balance between groups. For further details, see Supplementary Table S5.

**Comparison of statistical methods.** To systematically compare the estimated confounder-adjusted reductions, we proceeded as follows. We use Cohen's $d$[52] for descriptive statistics, the SMD[53] for PSM, and the average treatment effect as provided by causal forest. Since Cohen's $d$ and SMD are standardized measures, while the ATE is not, we re-transformed Cohen's $d$ and SMD into the same scale as the ATE to allow for comparability.

**Subgroup analysis.** To understand the variability in vitamin D levels across different subpopulations, we performed a subgroup analysis by evaluating the estimated changes within specific subgroups defined by age, gender, and season. First, we categorized the individuals into predefined subgroups: three age groups (i.e., 18–39, 40–59, 60+), two gender groups (i.e., women, men), and four season groups (i.e., winter, spring, summer, fall). We then ran a separate causal forest model for each subgroup. For each subgroup, we reported the estimated confounder-adjusted reduction in the outcome variable together with 99% and 95% confidence intervals (CI) and the percentage change relative to the average value across all individuals.

**Robustness checks.** To verify the robustness of our findings, we performed a series of robustness checks. First, we tested alternative time frames for the pre-pandemic and pandemic periods. The original analysis (March 2018–February 2020 vs. March 2020–February 2022) showed a mean vitamin D reduction of −0.7 µg/l and a deficiency rate increase of +4.0% (descriptive), with PSM-adjusted values of −0.5 µg/l and +3.6%. Adjusting the pre-pandemic period to end in January 2020 (i.e., March 2018–January 2020) and including February 2020 in the pandemic period yielded only a slightly larger difference (−0.9 µg/l, +4.9%), while PSM estimates remained unchanged (−0.5 µg/l, +3.6%).

Also, adjusting the pre-pandemic period to include March 2020 (i.e., March 2018–March 2020) resulted in consistent findings (−0.6 µg/l, +3.2%; PSM: −0.5 µg/l, +3.3%).

Second, we restricted the analysis to the months of March, April, and May in 2018, 2019, 2020, and 2021. This analysis confirmed the reduction in mean vitamin D levels (−2.3 µg/l) and the increase in deficiency rates (+4.3%). This is further corroborated by the PSM-adjusted estimates, which compute to −1.6 µg/l and +6.2%, respectively. When restricting the data to the above-defined months, the causal forest estimated a reduction of −2.042 µg/l, corresponding to an −8.1% decline during the pandemic compared to the pre-pandemic period. Together, the checks help rule out that the reductions rely on the specific choice of a month/season.

Third, we incorporated hours of sunshine (taken from[54]) as an additional covariate. Note that, due to the variation in the sunshine hours at the monthly level, it was not possible to perform PSM with exact matching, and, instead, we focused on the causal forest. The confounder-adjusted model estimated a reduction of −6.45 µg/l, indicating an even stronger decreasing in vitamin D levels during the pandemic, showing that the observed decline was not solely driven by seasonal sunlight variations.

Fourth, to test for unmeasured confounding, we conducted a negative control outcome analysis[55], which involves selecting an outcome that should not be affected by the intervention, given proper adjustment for confounders. Due to the absence of other variables that could act as such interventions, we randomly shuffled the outcome variable (i.e., vitamin D values) within the dataset and then repeated all analyses. As expected, this yielded no meaningful differences between the pre-pandemic and pandemic periods (descriptive: −0.1 µg/l, +0.1%; PSM: −0.1 µg/l, +0.1%; ML: ATE −0.0781 µg/l). Hence, the results reveal no significant association between intervention and negative outcome, indicating that unmeasured confounding was unlikely to bias the findings.

**Software.** All statistical analyses were performed using the statistical software R (version 4.4.1)[56]. For PSM, we use the `matchit` function from the package `MatchIt` (version 2.3.2)[57]. For the causal forest, we used the package `grf` (version 4.5.5)[51]. For the sensitivity analysis, we used the package `sensemakr` (version 0.1.6)[58].

### Ethics approval
The data request underwent a comprehensive assessment by an external compliance board, including patient and scientific representatives, to ensure the project met the ethical and legal standards required for using Honic data. This study received an Institutional Review Board exemption from the Ethics Committee of the Medical Faculty of LMU Munich (4-0900-KB).

### Reporting summary
Further information on research design is available in the Nature Portfolio Reporting Summary linked to this article.

## Data availability
The data used in this study were licensed from Health Data Technologies GmbH ("Honic") and are subject to contractual restrictions. The de-identified patient-level data cannot be shared due to patient privacy consideration and licensing agreements. In accordance with the license agreements, access may be granted for the purpose of reviewing the study within a secure analytics environment, interested researchers may request access by contacting Honic (https://honic.eu/en/contact-0). Requests will typically be reviewed within approximately 4–8 weeks. In accordance with data-sharing policies, only aggregated data can be made publicly available. For each figure, the corresponding aggregated data are provided in the source data file,

with patient-level data reported exclusively as summary statistics. Source data are provided with this paper.

## Code availability
The R code used for the main analysis is available at the public GitHub repository https://github.com/leaskapetze/vitamind_covid (https://doi.org/10.5281/zenodo.16980785).

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

## Acknowledgements

We thank our partner company Health Data Technologies GmbH ("Honic") for their collaboration and the expertise provided to the authors.

## Author contributions

L.S.: Study concept and design, data processing, statistical analysis, interpretation of the results, writing the manuscript. D.K.: Study concept and design, critical review of the manuscript. A.Z.: Critical revision of the manuscript. S.F.: Study concept and design, interpretation of the results, writing the manuscript. A.R.: Study concept and design, data processing, statistical analysis, interpretation of the results, writing the manuscript. E.G.: Study concept and design, interpretation of the results, writing the manuscript.

## Funding

## Competing interests

L.S. and A.R. are employed by Health Data Technologies GmbH ("Honic"), the data platform that provided access to the data used in this study. The remaining authors declare no competing interests.
