## [Transparent Peer Review file · Nature Communications]

Monitoring changes in vitamin D levels during the COVID-19 pandemic with routinely-collected laboratory data

Corresponding Author: Professor Eva Grill

Version 0:

Reviewer comments:

Reviewer #1

(Remarks to the Author)

Sample Selection and Potential Bias - The pandemic might have changed who comes for vitamin D testing. Public perceptions that vitamin D might influence COVID-19 outcomes could have drawn a different subset of people to the lab than in pre-pandemic times, thereby biasing the results. This is especially relevant given that your data are drawn from a patient/lab-based population rather than a randomized sample of the general population. Please discuss this limitation more explicitly in the manuscript.

Skewness of Vitamin D Distributions- Vitamin D data are typically skewed, as your Supplementary Figure S2 suggests. Although you present mean and standard deviation, it would be more statistically appropriate to report median and interquartile range (or first and third quartiles) and use nonparametric tests (e.g., Wilcoxon rank-sum) when comparing groups. At minimum, providing these additional metrics (medians, IQRs) would strengthen the robustness of the results in Table 2 and beyond.

Use of Regression Modeling- While the propensity score matching and causal forest approaches are commendable, many readers would also expect to see a simpler (but still powerful) regression approach—particularly, for example, a multivariable logistic regression—to adjust for confounders. Providing such an analysis would allow readers to directly compare effect estimates (e.g., odds ratios for vitamin D deficiency, or adjusted differences in vitamin D levels) and might enhance clarity for those less familiar with propensity scores or causal forests.

Potential for Longitudinal Analysis - You mention discarding individuals who had multiple vitamin D measurements. However, if there is a subset of patients with measurements in both the pre-pandemic and pandemic periods, analyzing these individuals longitudinally could yield a more powerful assessment of within-person changes over time. Even if the subset is smaller, a longitudinal analysis might provide additional evidence that would support or corroborate the cross-sectional findings.

Minor Comments- Ensure consistent usage of significant figures throughout the manuscript.

Addressing these points will further strengthen the manuscript's rigor and clarity. I look forward to seeing the revised version.

(Remarks on code availability)

-The code is well-organized but lacks a README.

-All required packages are listed at the start, but specifying version requirements or providing an environment lockfile would improve reproducibility.

-Users with the same data and package versions should replicate the analysis; a brief usage guide would make it more community-friendly.

-Overall, the code is valuable but would benefit from documentation ensuring fully reproducible workflows.

Reviewer #2

(Remarks to the Author)

This study used laboratory testing data to assess changes in vitamin D levels among nearly 300,000 patients from two years

pre-COVID to the first two years of the COVID pandemic. Given the importance of Vitamin D for a number of health outcomes, this is an important public health study. While portions of the manuscript are very strong (e.g., multiple analytic approaches), the manuscript could benefit by addressing the comments below.

Major

While the use of a population-wide dataset is valuable, the study is still limited by including both inpatient and outpatient data, given the large differences in the health status of individuals in these settings and the possibility that such testing changed over time. The site location should be added as a covariate in the adjusted and matched analyses; this is particularly important given the change in patient populations from pre- to post-COVID. Relatedly, some of the findings should better highlight that there could simply be changes in the underlying population (e.g., line 189 could be due to sicker young adults being in the healthcare system and getting a vitamin d test).

Similarly, it appears as if the study team had access to geographic information. Adjusting for the type of testing facility as well as the geographic location could further strength the causal approaches suggested in this study.

It is unclear why a linear time trend was not included to better account for changes in Vitamin D levels over time, particularly to account for any trends that occurred prior to the post-COVID date.

Relatedly, it would be helpful to better understand whether there were any differences at the early start of the pandemic, when quarantine was stricter versus the end of the pandemic and/or whether the impact of reduced Vitamin D exposure took time to accrue.

On page 12, please clarify whether differences in the changes were statistically tested. Given some of the text (e.g., Lines 93-94 and the associated p-value) it seems as if perhaps they were; however, the methods do not discuss how such differences were tested for. The methods only describe the change pre- to post-COVID for a specific group. This is particularly important given one of the study's main findings is that "older women had the largest decrease..." What is meant by "PSM adjusted analysis" (e.g., line 110). What type of model is this and what was adjusted for? The variables included in the PS match are clear in this section (line 108) and the methods section at the end; however, the type of model for the adjusted analysis itself as well as the covariates are not clear.

More information is needed regarding the outliers. What percent of records were above the 70 ug/l, and is this a clinically meaningful threshold? Are the findings robust to top coding at 70 ug/l?

The statistical analysis section notes an alpha of .05, but other thresholds are used throughout the paper, such as Figure S1 and Table S2.

What covariates are mentioned on line 361? Were the 3 covariates not exact matched on, thus having a standardized difference of 0.0?

Minor

The authors are applauded for generally very clear and helpful graphics, particularly Figures 1 and 2. Figure 3 could use a little cleaning up with respect to the absolute and relative differences labeling. The labeling in the header above each violin chart section suggests that both numbers are absolute changes.

Similarly, please provide additional information about what is shown in the violin plots to facilitate interpretation across readers. For example, Figure S2 states "mean" but there are not black dots. Typically the horizontal line in the box and whisker represents the median. What is represented by the whiskers (e.g., 2 SD, 95% CI, or something else)?

Please clarify whether "least decline" on line 139 (and similar phrasing in other places) are relative or absolute. Without referencing the model type could be confusing to readers. This should be clarified in Figure 5 as well.

Some places use "treatment" and "control," which could be switched to post and pre pandemic for clarity.

The phrasing used in line 185 is not exactly accurate, as it does not support the previous sentence directly. Furthermore, Table 1 does not show "lower test rates among younger adults." It shows that your sample had a lower percentage of younger adults relative to the percent that was older adults.

(Remarks on code availability)

Version 1:

Reviewer comments:

Reviewer #1

(Remarks to the Author)

Thank you for the thorough and thoughtful revision. The manuscript is substantially improved, and you have addressed my prior comments well. I have just one remaining clarity issue regarding Tables S7 and S8:

- Explicitly list all reference categories in the table itself. Winter, female, pre-pandemic, and age 18–39 should appear as rows marked “— (ref)”.
- Show every level for each categorical variable. If some levels are not present in the data, state this in a footnote. If the sample is restricted to adults, please note “Participants were ≥ 18 years; no < 18 category.”
- Label each non-reference level as a comparison. For example: “Male (vs Female)”, “Pandemic (vs Pre-pandemic)”, “Spring (vs Winter)”.

With these small edits, I'm satisfied.

Reviewer #2

(Remarks to the Author)

The authors have carefully addressed each of the comments.

The manuscript provides a robust analysis to assess the vitamin D levels among a large population before and after the COVID-19 pandemic.

Revised submission to *Nature Communications*

Manuscript: "Monitoring changes in vitamin D levels during the COVID-19 pandemic with routinely-collected laboratory data" (NCOMMS-25-08274)

Dear editor, dear reviewers,

We are deeply grateful for your helpful and detailed reviews on the previous version of our paper. We especially appreciate the constructive feedback regarding specific areas in need of improvement and have taken it very seriously. You will see that we have invested significant effort to bring the paper up to a substantially higher standard of quality by following your comments closely.

Based on your feedback, the **main improvements** to our manuscript are as follows:

1. **New logistic and linear regression analyses.** To strengthen the interpretation of our findings, we performed additional regression-based sensitivity analyses, including multivariate logistic regression to assess odds of vitamin D deficiency and linear regression to evaluate adjusted differences in continuous serum levels. These analyses complement our propensity score and causal forest methods, and confirm the robustness of our results. We provide the detailed results in our new Supplement B.
2. **New geographic analysis.** We have requested a new data package that included the 2-digit postal code as well as a regional variation mapping that was conducted by the provider using the 5-digit postal code. This enables us to stratify patients by region type (city, town, rural). More granular regional analyses (e.g., at the district or address level) were not feasible due to constraints imposed by the ethical oversight board, as they would increase the risk of re-identification. Nevertheless, our new analysis still offers granular, geographic insights to explore whether region type may explain our findings. We then repeated all three analyses (descriptive, propensity-score matched, and causal forest) with our new covariate to assess potential geographic heterogeneity in pandemic effects. Overall, we find qualitatively similar results. Our results are detailed in our new Supplement C.
3. **New longitudinal analysis.** We added a new analysis including a time trend with a continuous variable defined as the number of months before/after the onset of the pandemic (new Supplement D). In response to the suggestions from the reviewers, we further performed a new, within-subject longitudinal analysis on the subset of patients with repeated measurements both before and during the pandemic ($N = 61,393$). Specifically, we fitted a linear mixed-effects model to assess changes in vitamin D levels before and during the pandemic. The model included fixed effects for period, age group, gender, and season, and random intercepts and slopes for period at the patient level to account for repeated measurements. This new analysis revealed a statistically significant mean increase in vitamin D levels during the pandemic, which offers complementary insights into patient-level trends. Our new results and interpretation are presented in our new Supplement E.

We also summarize our new robustness checks in our main paper (see our new Section "Robustness checks"). Further, we followed all of your additional comments closely and provide detailed point-by-point responses below. Further, we highlighted all key changes in our manuscript in **blue text color**.

RESPONSES TO REVIEWER #1

Comment R1.1: *“Sample Selection and Potential Bias - The pandemic might have changed who comes for vitamin D testing. Public perceptions that vitamin D might influence COVID-19 outcomes could have drawn a different subset of people to the lab than in pre-pandemic times, thereby biasing the results. This is especially relevant given that your data are drawn from a patient/lab-based population rather than a randomized sample of the general population. Please discuss this limitation more explicitly in the manuscript.”*

Response R1.1: We thank the reviewer for this insightful and important comment. As a result, we improved our manuscript in two key ways:

First, we performed a new, within-subject longitudinal analysis on the subset of patients with repeated measurements both before and during the pandemic ($N = 61,393$). This analysis reveals new insights into patient-level vitamin D trends during the pandemic. We present the new results and interpretation in our new Supplement E.

Second, we followed your suggestion closely, and, as a result, we have added a dedicated paragraph in our revised Discussion section that explicitly discusses the above-mentioned potential bias. We now acknowledge more clearly that the observed differences in vitamin D levels across time periods may partially reflect changes in the underlying population presenting for testing—likely due to changes in public awareness, health-seeking behavior, or access to care—rather than solely true shifts in vitamin D status at the population level. In line with your comment, we specifically note that heightened public interest in vitamin D, due to its hypothesized role in mitigating COVID-19 risk, may have altered the demographic or clinical characteristics of individuals seeking testing during the pandemic compared to the pre-pandemic period. While we attempted to mitigate such differences through careful matching and stratification approaches (e.g., propensity score matching, seasonal adjustment, and subgroup analyses), we recognize that these methods cannot fully eliminate biases introduced by shifts in testing behavior over time. We therefore further emphasize that future studies leveraging population-representative data sources or random sampling approaches would be valuable in validating and extending our findings.

Comment R1.2: *“Skewness of Vitamin D Distributions- Vitamin D data are typically skewed, as your Supplementary Figure S2 suggests. Although you present mean and standard deviation, it would be more statistically appropriate to report median and interquartile range (or first and third quartiles) and use nonparametric tests (e.g., Wilcoxon rank-sum) when comparing groups. At minimum, providing these additional metrics (medians, IQRs) would strengthen the robustness of the results in Table 2 and beyond.”*

Response R1.2: Thank you! In response to your suggestion, we have added a new table (new Supplementary Table S1) where we present the median values and interquartile ranges for all subgroup comparisons (age groups, gender, and season), alongside the mean and standard deviation values.

Comment R1.3: *“Use of Regression Modeling- While the propensity score matching and causal forest approaches are commendable, many readers would also expect to see a simpler (but still powerful) regression approach—particularly, for example, a multivariable logistic regression—to adjust for confounders. Providing such an analysis would allow readers to directly compare effect estimates (e.g., odds ratios for vitamin D deficiency, or adjusted differences in vitamin D levels) and might enhance clarity for those less familiar with propensity scores or causal forests.”*

Response R1.3: Thank you for your thoughtful suggestion. In response, we have conducted two additional regression-based sensitivity analyses to strengthen our analysis. Specifically, we performed (1) a multivariable logistic regression to assess the adjusted odds of vitamin D deficiency, and (2) a multivariable linear regression to estimate adjusted differences in continuous serum vitamin D levels. These analyses confirm the robustness of our findings, particularly the association between the pandemic period and vitamin D outcomes. Importantly, the results are fully consistent with those obtained through the propensity score and causal forest approaches. We have added the new results to the Supplementary Material; see new Supplement B and the tables therein.

Comment R1.4: *“Potential for Longitudinal Analysis - You mention discarding individuals who had multiple vitamin D measurements. However, if there is a subset of patients with measurements in both the pre-pandemic and pandemic periods, analyzing these individuals longitudinally could yield a more powerful assessment of within-person changes over time. Even if the subset is smaller, a longitudinal analysis might provide additional evidence that would support or corroborate the cross-sectional findings.”*

Response R1.4: Thank you for your thoughtful feedback. We thus followed your suggestion closely and performed a new longitudinal within-subject analysis where we focus on the subset of patients who had at least one vitamin D measurement both before and during the pandemic. This analysis includes 61,393 patients with repeated measurements. Specifically, we fitted a linear mixed-effects model to assess changes in vitamin D levels before and during the pandemic. The model included fixed effects for period, age group, gender, and season, and random intercepts and slopes for period at the patient level to account for repeated measurements. The results showed a significant increase in vitamin D levels during the pandemic compared to the pre-pandemic period. We report the results in our new Supplement E. The results offer an important complementary perspective to our cross-sectional findings. While our main analysis showed a population-level decrease in vitamin D levels during the pandemic, this within-subject analysis may reflect improved follow-up care or targeted interventions (e.g., supplementation) in the subpopulation of patients with multiple measurements, and thereby provides additional nuance to the overall interpretation.

Comment R1.5: *“Minor Comments- Ensure consistent usage of significant figures throughout the manuscript”*

Response R1.5: Thanks! We have carefully revised the manuscript to ensure consistent usage of significant figures.

Comment R1.6: *“Addressing these points will further strengthen the manuscript’s rigor and clarity. I look forward to seeing the revised version.”*

Response R1.6: Thanks for seeing value in our manuscript. We are deeply grateful for your thoughtful feedback, which has been a great help in improving the rigor and generalizability of our study and thereby strengthening the evidence presented. We hope you agree.

Comment R1.7: *“Reviewer #1 (Remarks on code availability):*

-The code is well-organized but lacks a README.

-All required packages are listed at the start, but specifying version requirements or providing an environment lockfile would improve reproducibility.

-Users with the same data and package versions should replicate the analysis; a brief usage guide would make it more community-friendly.

-Overall, the code is valuable but would benefit from documentation ensuring fully reproducible workflows.”

Response R1.7: We deeply appreciate your suggestion. We have created a detailed README file that now includes a brief usage guide and clear instructions for running the code. In addition, to improve reproducibility, we generated and attached a `renv.lock` file specifying all package versions used in the analysis, ensuring that users with the same data can fully replicate our results (see our revised Supplementary Files).

RESPONSES TO REVIEWER #2

Comment R2.0: *“This study used laboratory testing data to assess changes in vitamin D levels among nearly 300,000 patients from two years pre-COVID to the first two years of the COVID pandemic. Given the importance of Vitamin D for a number of health outcomes, this is an important public health study. While portions of the manuscript are very strong (e.g., multiple analytic approaches), the manuscript could benefit by addressing the comments below.”*

Response R2.0: We thank you for your thoughtful and encouraging feedback. We are delighted that you find our study important and overall very strong with regard to the multiple analytic approaches we employed. We are also grateful for the constructive comments, which have carefully addressed each point to further strengthen our manuscript. Thank you!

Major

Comment R2.1: *“While the use of a population-wide dataset is valuable, the study is still limited by including both inpatient and outpatient data, given the large differences in the health status of individuals in these settings and the possibility that such testing changed over time. The site location should be added as a covariate in the adjusted and matched analyses; this is particularly important given the change in patient populations from pre- to post-COVID. Relatedly, some of the findings should better highlight that there could simply be changes in the underlying population (e.g., line 189 could be due to sicker young adults being in the healthcare system and getting a vitamin d test).”*

Response R2.1: Thank you for your feedback. We followed your suggestions closely and, as a result, improved our manuscript in the following ways.

First, we have now included a new analysis that includes the 2-digit postal code as an additional covariate in the adjusted and matched analyses to account for potential regional differences and shifts in patient populations over time. Furthermore, the data provider was able to map the 5-digit postal codes to their corresponding types of area (city, town, rural, etc.) and this allowed us to conduct additional analyses on the changes in vitamin D mean values and deficiency rates depending on the types of areas. However, it was not possible to address this topic by using the facility’s location but only using the patient’s location. Nevertheless, we believe the additions we could make significantly strengthen the robustness of our analyses. We added the results of our new analyses to the supplementary material (see new Supplement C). Thank you for this important and very helpful point!

Second, we would like to highlight that we conducted propensity score matching with exact matching simultaneously on *both* age groups *and* gender, which allows us to directly account for changes in the population composition between the pre- and post-COVID periods. To illustrate this, we now cross-reference Supplementary Table S2 which shows the characteristics of the matched population, and, further, we have added a new Supplementary Table S1 where we offer quantitative comparisons for the observed population. Upon reading your later comments, we realized that we should have been more explicit regarding which variables we adjusted for in the propensity score matching analysis, and we now state this more explicitly in the main paper (i.e., age, gender, and season). Hence, we essentially control for the fact that younger adults may have a different propensity to getting a vitamin D test, but we cannot control for their overall health state. To address the latter point, we now better

highlight that, while we control for several characteristics of the patient population, there may be other, unobserved changes in the underlying population. In particular, we revised the above-mentioned statement and spelled out our limitations clearly in that we estimated confounder-adjusted changes across, e.g., gender, age, and region, but that there may be other factors that can explain some of the observed trends (see our revised Discussion section).

Comment R2.2: *“Similarly, it appears as if the study team had access to geographic information. Adjusting for the type of testing facility as well as the geographic location could further strength the causal approaches suggested in this study.”*

Response R2.2: We thank the reviewer for this valuable suggestion. While we do not have access to information about the specific type of testing facility, we were able to revisit our database and retrieve information on the geographical origin of the study population. Specifically, the data provider could map the 5-digit postal code to three categories that reflect the type of area: city, town/suburb, and rural area. Please note that more granular regional analyses (e.g., at the district or address level) were not feasible due to constraints imposed by the ethical oversight board, as they would increase the risk of re-identification.

Still, based on this newly added geographic covariate, we expanded our analyses in three ways: (1) We repeated the descriptive analyses to compare vitamin D levels and deficiency rates across the three area types before and during the pandemic. (2) We repeated the propensity score matching while adjusting for the geographic variable and report updated effect sizes. (3) We included the regional covariate in our causal forest analysis to explore potential heterogeneous treatment effects. This allowed us to assess whether the estimated pandemic effect on vitamin D levels differed by geographic region. Overall, the results are qualitatively similar.

We provide the new results in our new Supplement C. Therein, we now include two new tables summarizing vitamin D levels and deficiency rates by type of region, as well as an additional plot that displays the region-specific average treatment effects derived from the causal forest model.

Comment R2.3: *“It is unclear why a linear time trend was not included to better account for changes in Vitamin D levels over time, particularly to account for any trends that occurred prior to the post-COVID date.”*

Relatedly, it would be helpful to better understand whether there were any differences at the early start of the pandemic, when quarantine was stricter versus the end of the pandemic and/or whether the impact of reduced Vitamin D exposure took time to accrue.”

Response R2.3: Thank you very much for this helpful comment. We followed suggestions and now improved our manuscript in the following three ways:

First, we have now added two supplementary figures to address these points. Our Supplementary Figure S1 shows the mean vitamin D levels over time across seasons from Winter 2018 to Winter 2022, allowing visualization of any gradual pre-pandemic or post-pandemic trends. Our new Supplementary Figure S2 additionally compares vitamin D levels between the early (March–August 2020) and late (September 2021–February 2022) pandemic periods, to assess potential differences between the beginning and later phases of the pandemic.

Second, we performed the suggested analysis including a time trend. Specifically, we included a continuous time trend variable, defined as the number of months before or after the onset of the pandemic. The new results are in our new Supplement D.

Third, we performed a new, within-subject longitudinal analysis on the subset of patients with repeated measurements both before and during the pandemic ($N = 61,393$). This new analysis revealed a statistically significant mean increase in vitamin D levels during the pandemic, which offers complementary insights into patient-level trends. Our new results and interpretation are presented in our new Supplement E.

Comment R2.4: *“On page 12, please clarify whether differences in the changes were statistically tested. Given some of the text (e.g., Lines 93-94 and the associated p-value) it seems as if perhaps they were; however, the methods do not discuss how such differences were tested for. The methods only describe the change pre- to post-COVID for a specific group. This is particularly important given one of the study’s main findings is that “older women had the largest decrease...”*

What is meant by “PSM adjusted analysis” (e.g., line 110). What type of model is this and what was adjusted for? The variables included in the PS match are clear in this section (line 108) and the methods section at the end; however, the type of model for the adjusted analysis itself as well as the covariates are not clear..”

Response R2.4: Thank you very much for this important comment. We apologize for not phrasing this more clearly in the original submission. To address your comment, we revised our manuscript as follows. (1) We have now clarified in the Methods section that differences in vitamin D levels and deficiency rates between groups were statistically tested using *t*-tests and chi-square tests (see revised Methods), both in the unmatched and the matched samples. Overall, we were cautious about relying solely on *p*-values in our discussion given the large sample size, and therefore emphasized the estimated differences in the interpretation of our results. (2) We have also added a detailed description specifying that the "PSM-adjusted analysis" refers to direct comparisons of outcomes within the matched sample without fitting an additional regression model. These clarifications have been incorporated to ensure full transparency regarding our analytic approach. (3) We state explicitly in the main part of the paper that the PSM-adjusted analysis accounts simultaneously for age, gender, and season.

Comment R2.5: *“More information is needed regarding the outliers. What percent of records were above the 70 ug/l, and is this a clinically meaningful threshold? Are the findings robust to top coding at 70 ug/l?”*

Response R2.5: Thank you very much for raising this important point. We apologize for not offering a more detailed explanation in the original version. As we now clarify in the revised manuscript, values above 70 $\mu\text{g/l}$ (0.98% of all records) were removed solely for visualization purposes to prevent extreme values from distorting the scale of the plots. All statistical analyses, including group means and significance tests, were conducted on the full dataset without the removal of any high values to ensure representativeness. The threshold of 70 $\mu\text{g/l}$ was chosen based on clinical guidelines indicating that serum 25(OH)D concentrations above 50–70 $\mu\text{g/l}$ are considered unusually high (Holick et al., 2011).

Comment R2.6: *“The statistical analysis section notes an alpha of .05, but other thresholds are used throughout the paper, such as Figure S1 and Table S2.”*

Response R2.6: Thank you. We have now streamlined the alpha and now use consistently the stricter significance level of $\alpha = 0.01$ throughout the manuscript, figures, and tables. We have updated the Methods section accordingly and ensured that all significance interpretations and thresholds consistently reflect this standard. We chose a stricter alpha level of 0.01 given the very large sample size of our study.

Comment R2.7: *“What covariates are mentioned on line 361? Were the 3 covariates not exact matched on, thus having a standardized difference of 0.0?”*

Response R2.7: We thank the reviewer for this valuable comment and, upon reading the comment, realized that the text needs clarification. In our initial analysis (reported in the main manuscript), the covariates age group, gender, and test month were included in the propensity score model and balanced using inverse probability of treatment weighting (IPTW), rather than exact matching. The previously reported weighted standardized mean differences (ASMDs) of 0.000 for these variables referred to the mean ASMD calculated across all levels of each covariate.

However, we acknowledge that this may have been misleading, as two very different categorical distributions can still yield a near-zero mean ASMD. To provide a more granular view of balance, we now report in the new Supplementary Table S6 the weighted ASMDs for each individual category of age group, gender, and test month (e.g., age group 18–39, test month April). These values were all well below the commonly accepted threshold of 0.1, confirming adequate covariate balance at the category level.

In addition, we calculated the variance ratio (treated vs. control) of the estimated propensity scores, which was 0.935, falling within the recommended range of 0.5–2.0 for acceptable balance. This provides further assurance that covariate distributions were sufficiently balanced between groups after weighting.

To improve our manuscript, we have clarified this point and added a new Supplementary Table S6 to reflect the full categorical breakdown and variance ratio as part of our balance diagnostics.

Minor

Comment R2.8: *“The authors are applauded for generally very clear and helpful graphics, particularly Figures 1 and 2. Figure 3 could use a little cleaning up with respect to the absolute and relative differences labeling. The labeling in the header above each violin chart section suggests that both numbers are absolute changes?”*

Response R2.8: Thank you for the suggestions to improve Figure 3. We have revised the labeling above each violin plot to clearly distinguish between absolute and relative changes, now stating "Abs." and "Rel." separately to avoid any potential confusion.

Comment R2.9: *“Similarly, please provide additional information about what is shown in the violin plots to facilitate interpretation across readers. For example, Figure S2 states “mean” but there are not black dots. Typically the horizontal line in the box and whisker represents the median. What is represented by the whiskers (e.g., 2 SD, 95% CI, or something else)?”*

Response R2.9: Thank you. We apologize for the lack of clarity in the original figure legend. We have now updated the caption for the Supplementary Figure S2 (now: Supplementary Figure S3) to explicitly state that the violin plots display the full distribution, the black box shows the interquartile range (IQR), the horizontal line represents the median, the whiskers extend to 1.5 times the IQR, and the black point and error bars represent the mean and one standard deviation (SD).

Comment R2.10: *“Please clarify whether “least decline” on line 139 (and similar phrasing in other places) are relative or absolute. Without referencing the model type could be confusing to readers. This should be clarified in Figure 5 as well.”*

Response R2.10: Thank you. We have clarified that all reported declines and subgroup effects represent absolute changes in mean vitamin D levels (measured in $\mu\text{g/l}$), unless otherwise specified. To improve clarity, we updated the caption of Figure 5 and added an explicit statement in the Methods section. We hope the changes make the results more intuitive for the reader.

Comment R2.11: *“Some places use “treatment” and “control,” which could be switched to post and pre pandemic for clarity.”*

Response R2.11: Thank you very much for this suggestion. We have carefully reviewed the manuscript and replaced "treatment" and "control" with "pandemic" and "pre-pandemic" terminology throughout the text to improve the clarity for the reader.

Comment R2.12: *“The phrasing used in line 185 is not exactly accurate, as it does not support the previous sentence directly. Furthermore, Table 1 does not show “lower test rates among younger adults.” It shows that your sample had a lower percentage of younger adults relative to the percent that was older adults.”*

Response R2.12: Thank you. We have revised the phrasing. To correctly reflect Table 1, we now state that the table shows a lower proportion of younger adults among those tested, rather than lower test rates per se (see our revised Discussion section).

References

Holick, M. F., Binkley, N. C., Bischoff-Ferrari, H. A., Gordon, C. M., Hanley, D. A., Heaney, R. P., ... & Weaver, C. M. (2011). Evaluation, Treatment, and Prevention of Vitamin D Deficiency: an Endocrine Society Clinical Practice Guideline. *The Journal of Clinical Endocrinology & Metabolism*, 96(7), 1911–1930.

NCOMMS-25-08274A

Response to reviewers

REVIEWERS' COMMENTS

Reviewer #1 (Remarks to the Author):

Thank you for the thorough and thoughtful revision. The manuscript is substantially improved, and you have addressed my prior comments well. I have just one remaining clarity issue regarding Tables S7 and S8:

- Explicitly list all reference categories in the table itself. Winter, female, pre-pandemic, and age 18–39 should appear as rows marked “— (ref)”.
- Show every level for each categorical variable. If some levels are not present in the data, state this in a footnote. If the sample is restricted to adults, please note “Participants were ≥ 18 years; no < 18 category.”
- Label each non-reference level as a comparison. For example: “Male (vs Female)”, “Pandemic (vs Pre-pandemic)”, “Spring (vs Winter)”.

With these small edits, I’m satisfied.

Reviewer #2 (Remarks to the Author):

The authors have carefully addressed each of the comments.

The manuscript provides a robust analysis to assess the vitamin D levels among a large population before and after the COVID-19 pandemic.

Thank you again for reviewing the manuscript. We have modified Tables S7 and S8 accordingly.